# Antioxidant Potential of Physicochemically Characterized *Gracilaria blodgettii* Sulfated Polysaccharides

**DOI:** 10.3390/polym13030442

**Published:** 2021-01-30

**Authors:** Bilal Muhammad Khan, Li-Xin Zheng, Wajid Khan, Aftab Ali Shah, Yang Liu, Kit-Leong Cheong

**Affiliations:** 1Guangdong Provincial Key Laboratory of Marine Biotechnology, STU-UNIVPM Joint Algal Research Center, Department of Biology, College of Science, Shantou University, Shantou 515063, China; dr.bilal@uaar.edu.pk (B.M.K.); 19lxzheng@stu.edu.cn (L.-X.Z.); liuyanglft@stu.edu.cn (Y.L.); 2University Institute of Biochemistry and Biotechnology, Pir Mehr Ali Shah Arid Agriculture University, Rawalpindi 46300, Punjab, Pakistan; 3National Center of Industrial Biotechnology, Pir Mehr Ali Shah Arid Agriculture University, Rawalpindi 46300, Punjab, Pakistan; 4Center for Biotechnology and Microbiology, University of Swat, Mingora 19130, Khyber Pakhtunkhwa, Pakistan; sherafghan.shah@gmail.com; 5Department of Biotechnology, Faculty of Biological Sciences, University of Malakand, Chakdara, Lower Dir 18800, Khyber Pakhtunkhwa, Pakistan; aftabuom@gmail.com

**Keywords:** red algal, *Gracilaria blodgettii*, polysaccharide, antioxidant

## Abstract

Marine rhodophyte polysaccharides have a wide range of described biological properties with nontoxic characteristics, and show great potential in prebiotics and the functional foods industries. However, there is a virtual lack of *Gracilaria blodgettii* polysaccharides (GBP) profiling and their bioactivities. This study was designed while keeping in view the lack of physical and chemical characterization of GBP. This polysaccharide was also not previously tested for any bioactivities. A linear random coil conformation was observed for GBP, which was found to be a polysaccharide. A significant sulfate (*w/w*, 9.16%) and 3,6-anhydrogalactose (AHG, *w/w*, 17.97%) content was found in GBP. The significant difference in its setting (27.33 °C) and melting (64.33 °C) points makes it resistant to increasing heat. This, in turn, points to its utility in industrial scale processing and in enhancing the shelf-life of products under high temperatures. A radical scavenging activity of 19.80%, 25.42% and 8.80% was noted for GBP (3 mg/mL) in 2,2-diphenyl-1-picrylhydrazyl (DPPH), 2,2’-azino-bis (ABTS) and hydroxyl radical (HO) scavenging assays, respectively. Therefore, the findings suggest that *Gracilaria blodgettii* polysaccharides display a good antioxidant potential and may have potential applications in the functional food industry.

## 1. Introduction

Covering the majority of the earth’s surface, oceans are inhabited by one of the most abundant, diverse and nutritionally important group of organisms [1]. As one of the most abundant organisms, algae has aroused an increasing interest in consumption all around the world, being used in the food, cosmetic or pharmaceutical industries [2,3,4]. Over the past 20 years, the global market of seaweed consumption has grown by about 20% [5]. Based on the different colors of thalli, algae are classified into three types: Rhodophyceae (red algae), Phaeophyceae (brown algae) and Chlorophyceae (green algae) [6]. Marine Rhodophytes contain distinctive polysaccharides, which are predominantly agarans or carrageenans [7]. In recent years, polysaccharides extracted from marine algae have been reported to exhibit various biological activities such as antioxidant, anticoagulant, antitumor, anti-inflammatory and immunomodulatory effects [8,9,10]. They have a wide range of health benefits for humans, including anti-aging, anti-breast cancer, gastroprotective and neuroprotection effects [8]. Thus, they became more and more important in biochemical and medical areas, also having received increasing attention in the past decade [11].

Rhodophytal polysaccharides’ backbone consists of a repetitive 3-linked β-D-galactopyranose and 4-linked α-galactopyranose disaccharide unit [12]. The carrageenan and agar of red algae are mainly composed of galactans [13,14]. Their α-galactose residues, on the other hand, are either primarily or entirely comprised of 3,6-anhydrogalactose [15].

Red algae, the profuse and primordial marine macrophytes [16], comprises one of the largest phyla of algae. It belongs to the genus *Gracilaria*, and it evidently shows its importance in the form of agrophytes by abundantly occupying vegetable market shelves as salad and serves as an imperative ingredient in the production of nonalcoholic drinks and certain foods [17]. It can grow and reproduce in a vast habitat extending from the Arctic Ocean to the Northern Hemisphere’s tropical seas, where Southeast Asia is located [18]. Given its easy accessibility, the genus *Gracilaria* is important to industries as the main source of agar, and it is also consumed locally as an everyday food with great demand [17]. In addition, it also has the potential to be used in the manufacturing of functional foods and nutraceuticals [19,20], which is a vibrant revenue-making industrial segment [21].

Reactive oxygen species (ROS) can be manifested in the form of apoptosis, cardiovascular disease and even cancer. Meanwhile, antioxidants combat the harmful effects of ROS, like the degenerative or pathological processes of various serious diseases such as inflammation, cardiac and neurological conditions, and nucleic acid, protein and lipid deformations [22,23,24]. The antioxidant potency of various seaweed-extracted sulfated polysaccharides has been previously reported. For instance, Patel reported that the sulfated polysaccharide extracted from red algae exhibited strong antioxidant, viz. carrageenans [25], while Bhadja et al. also found that polysaccharides with a high sulfate content significantly increased their antioxidant activity [8]. Furthermore, Torres et al. also reported that antioxidant activity is one of the most evaluated properties of the genus *Gracilaria* [26]. In general, the biological activities of SP are strongly related to their chemical structure, such as the content of sulfate groups and uronic acid, as well as its molecular weight [27]. *Gracilaria blodgettii*, however, virtually lacks specific polysaccharide profiling and bioactivities. Therefore, the evaluation and comparison of the polysaccharides of *Gracilaria blodgettii*, considering their chemical composition and bioactivities, were particularly necessary to broaden their applications in functional foods industry. The aim of the present study was to emphasize the physical and chemical characterization of *Gracilaria blodgettii* polysaccharides (GBP) and their potential utility as antioxidants. Our results may help to display potential applications in the functional food industry and might be useful for further biochemical studies in the future.

## 2. Materials and Methods

### 2.1. Plant Material and Reagents

After collecting *G. blodgettii* (Hainan Island, China) in December 2017, it was dried under vacuum conditions. The dried samples were ground into fine powder by a powder machine and sifted through an 80-mesh sieve. Aladdin (Shanghai China) provided Trifluoroacetic acid (TFA), acetonitrile and salicylate, while Sigma-Aldrich (St. Louis, MO, USA) delivered L-Ascorbic acid, 2,2-diphenyl-1-picrylhydrazyl (DPPH), 2,2’-azino-bis (ABTS) and 4-methyl-morpholine borane (MMB). Acetic anhydride, ethyl acetate and dichloromethane were procured from XiLong Scientific Co., China, and 3,6-anhydrogalactose (AHG) was procured from J&K Scientific Ltd., Beijing, China.

### 2.2. Polysaccharide Extraction

The removal of low molecular weight impurities was achieved by treating powdered material (25 g) with methanol/dichloromethane/water (4:2:1, *v/v/v*, 250 mL), and afterwards subjecting the resultant mixture to shaking incubation for 24 h at ambient temperature. This was followed by centrifugation (4000× *g*, 15 min) and the subsequent drying of the precipitate at 60 °C. The dried precipitate was resuspended in deionized distilled water (1:30, *w/v*), kept for 30 min at room temperature, and was then heat-treated in a water bath (90 °C) for 2 h. Following another round of centrifugation (4000× *g*, 15 min), 95% ethanol (three-fold) was added to the precipitate. The precipitate that was dissolved in ethanol was stored overnight at 4 °C before undergoing yet another round of centrifugation (4000× *g*, 15 min). The precipitate hence obtained was dissolved in hot water prior to freeze-drying for 24 h.

### 2.3. GBP Characterization

#### 2.3.1. Sulfate Content

A modified BaCl_2_-gelatin turbidimetric technique [28] was used for the sulfate content determination. In a capped vial, the polysaccharide weighing 5 mg was hydrolyzed with 1 mol/L HCl (2.5 mL) for 3 h at 100 °C. The hydrolysate was allowed to cool down and was then diluted to a final volume of 50 mL with 1 mol/L HCl. Afterwards, it was filtered and trichloroacetic acid (TCA) was added (3%; *v/v*; 3.8 mL), together with BaCl_2_-gelatin solution (1 mL). The uniformity of the solution was ensured through vigorous mixing. A settling time of 15 min was given at room temperature. It was followed by a measurement of the absorbance at 360 nm for this solution (A_1_) and a solution of GBP/gelatin/TCA (A_2_). Subsequently, the value of A_2_ was subtracted from the value obtained for A_1_, which gave us the final absorbance (GBPabs). The standard curve was generated using a solution of K_2_SO_4_.

#### 2.3.2. Total Sugar Content

The total sugar content of GBP was determined using the phenol-sulfuric acid method, where 200 μL of phenol solution (5%; *w/v*) and 1 mL of concentrated sulfuric acid was added to 200 μL of both the sample (2.5 mg/mL) and the standard glucose (0.2, 0.4, 0.6, 0.8 and 1 mg/mL). The mixtures, after vigorous shaking, were placed for 15 min in a boiling water bath and were then allowed to cool down. The measurement of the absorbance for the mixtures was carried out (490 nm) and was compared to the standard curve generated with glucose.

#### 2.3.3. Setting/Melting Points Determination

A thermometer was inserted into a test tube containing 1.5% GBP solution (8 mL) in order to determine its setting point. The test tube was placed in an ice bath to lower the temperature. The temperature at which the GBP solution stopped flowing even after tilting the test tube at an angle of 90 °C was recorded as the setting point. The melting point, on the other hand, was determined by placing a small metal ball on the surface of the solidified GBP solution. The GBP solution was transferred to a water bath, and the temperature was gradually increased. That temperature was noted as the melting point at which the metal ball touched the bottom of the test tube.

#### 2.3.4. High-Performance Size-Exclusion Chromatography

The weight-average molecular weight (*M*_w_), number-average molecular weight (*M*_n_), polydispersity (*M*_w_ /*M*_n_), PD, intrinsic viscosity [*η*], hydrodynamic radius (*R*_h_) and radius of gyration (*R*_g_) of GBP were determined using a high-performance size-exclusion chromatography with multiple detectors, a viscometer, a refractive index detector and multi-angle laser light scattering (HPSEC-Visc-MALLS-RID). An Agilent 1100 series LC/DAD (Liquid chromatography coupled to diode array detector) system connected with a PL aquagel-OH MIXED-H column (300 mm × 7.8 mm) was used. A 30 °C column temperature was maintained. The determination of [*η*] for GBP followed the classic Huggins–Kraemer equations. RID was calibrated using a series of aqueous NaCl solutions of known concentrations. The mobile phase, 0.1 mol/L NaNO_3_ aqueous solution, had a flow rate of 0.5 mL/min. All samples were analyzed in a fixed injection volume (50 μL). ASTRA 5.0 software (Wyatt Technology Co., Goleta, CA, USA) was employed for data collection and analyses.

#### 2.3.5. Reductive Hydrolysis and Acetylation

GCP was exposed to a twostep reductive hydrolysis with 4-methyl-morpholine borane (MMB) and trifluoroacetic acid (TFA), and afterwards to acetylation with ethyl acetate in accordance with a published protocol [29]. A Gas Chromatography-Mass Spectrometer (GC–MS) apparatus having an Rtx-5MS column (30 m × 0.2 mm × 0.25 µm) was used for AHG quantification in GBP. The injector temperature was fixed at 280 °C preceding the injection of the samples (1 μL with a split ratio of 1:10). The detector temperature was retained at 250 °C. The oven temperature regime was 60 °C (2 min) with an incremental rise of 30 °C/min up to 120 °C (1 min hold), and afterwards of 25 °C/min up to 250 °C (30 min hold). The carrier gas, N_2_, had a flow rate of 2 mL/min. Mass spectrometry was executed at 70 eV and 50–800 m/z scan fragments. The calibration curve for the quantification was obtained with standard AHG.

#### 2.3.6. Scanning Electron Microscopy (SEM)

The surface characterization and microstructure evaluation of GBP was carried out through SEM (JSM-6360LA; JEOL, Tokyo, Japan). Lyophilized GBP was coated with a thin layer of Au and was then evaluated, under vacuum, at an acceleration voltage of 10 kV.

#### 2.3.7. Atomic Force Microscopy (AFM)

GBP was also subjected to atomic force microscopic analysis using a scanning probe microscope (Dimension Icon; Bruker, Germany). GBP dissolved in distilled water at a concentration of 1 µg/mL was sonicated (30 min) and was then pipetted (2–3 µL) onto a mica disc. Afterwards, the disc was dried (120 °C for 30 s) before analysis.

#### 2.3.8. Fourier Transform Infrared Spectroscopy (FT-IR)

After pressing 1 mg of GBP and 100 mg KBr into a pellet, the FT–IR spectrum (MAGNA-IR 750, Thermo Nicolet Co., Waltham, Massachusetts USA) was measured in a fixed frequency range (4000–400 cm^−1^). The number of scans used was 32, and the spectral resolution was 2 cm^−1^.

### 2.4. Antioxidant Potential of GBP

#### 2.4.1. DPPH Radical Scavenging Activity

A previous publication [30], incorporating some modifications, was referred to in order to determine the DPPH radical scavenging potency of GBP. In brief, 0.004% (*w/v*) DPPH/dehydrated alcohol solution (1 mL) was well mixed with 1 mL GBP (0.25, 0.5, 1, 2 and 4 mg/mL). The reaction mixtures were allowed to react, in the dark, at room temperature for half an hour. Afterwards, the absorbance was measured at 517 nm, while the DPPH radical scavenging effect (%) was derived from a previously reported equation [31].

#### 2.4.2. Hydroxyl Radical (HO) Scavenging Activity

To evaluate the HO scavenging efficacy of GBP, a few amendments were made in a published protocol [32]. In brief, 1 mL each of GBP (0.5, 1, 2, 4 and 8 mg/mL), FeSO_4_ (9 mmol/L) and ethanol salicylate (9 mmol/L) were mixed to form a 1:1:1 solution. Afterwards, 1 mL of 9 mmol/L H_2_O_2_ was introduced preceding incubation (30 min; 37 °C). The absorbance was measured at 510 nm, while the HO scavenging activity (%) was calculated in accordance with a previous report [31].

#### 2.4.3. ABTS Radical Scavenging Activity

GBP was also ascertained for the presence of any ABTS radical scavenging effect according to a published protocol [33], with some alterations. To generate ABTS radicals, 5 mL of 7 mmol/L ABTS was mixed with 88 μL of 149 mmol/L K_2_S_2_O_8_. The resultant reaction mixture was subjected to incubation (dark; 16 h) prior to the addition of this solution (1 mL) to 10 μL of each GBP concentration (0.25, 0.5, 1, 2 and 4 mg/mL). The ABTS radical scavenging activity (%) was determined [34] after measuring the absorbance at 734 nm.

### 2.5. Statistical Analysis

Each experiment was repeated three times. The values were calculated through Microsoft Excel 2010 and were indicated as the mean ± standard deviation (where relevant).

## 3. Results and Discussion

### 3.1. Physical and Chemical Characteristics

The physical and chemical characteristics of GBP are summarized in Table 1. Showing consistency with the results communicated for other species of the genus *Gracilaria* [19,35], a yield of 24.12% was obtained for GBP. The range of the GBP sulfate content (9.16%) showed a similarity to that reported for *G. fisheri* [36], *G. chouae* [19] and *G. intermedia* [37]. The food, cosmetic and pharmaceutical applications of sulfated polysaccharides extracted from seaweed have been in the limelight during the previous few decades [38]. The total sugar content of GBP (18.84%), however, was quite lower than that reported for *G. fisheri* (56.33 and 55.63%) [36] and *G. chouae* (52.63%) [19].

The enormous dissimilarity between the observed setting and melting point for GBP (27.33 °C and 64.33 °C, respectively) corresponds to its good applicability in industrial processes. Being farther apart, such a setting and melting point allow for a high stability amidst increased temperatures during industrial processing [19]. Moreover, a higher melting point also corresponds to a greater utility in hot environments and in baked stuff, which provides more possibilities for GBP when cooking daily diets and in processing and applications within the food industry [39].

Several important indexes for polysaccharide characterization can be obtained from HPSEC-Visc-MALLS-RID analyses. These include *M*_w_ and *M*_n_, which correspond to the polymeric chain size and chain length, respectively. The polydispersity index (PDI) is another important parameter that is essentially the ratio of *M*_w_ to *M*_n_ (*M*_w_*/M*_n_). Yet another important index is represented by the intrinsic viscosity [*η*], an indication of the solute’s impact on the solution’s viscosity. This index is reportedly related to the polysaccharide’s configuration, its degree of branching and *M*_w_, and the property of the solvent [40]. Similarly, the Mark–Houwink–Sakurada α, *R*_g_ and *R*_h_ values are significant in revealing vital information about the polysaccharide’s structural conformation. GBP showed an *M*_w_ of 6.149 × 10^5^ Da and *M*_n_ of 4.169 × 10^5^ Da, indicating that GBP was a polysaccharide of natural origin [41]. A typical symmetric peak showing a slightly wider molecular mass distribution was obtained for GBP (Figure 1), testifying to the degree of dispersion present in it. The GBP was further evident from the observed PDI (1.475). Likewise, an 844.12 mL/g [*η*] value was evident for GBP. However, low [*η*] values are generally regarded as highly fermentable [42], but this interpretation is now changing [43]. This evidence has been specifically reported for algal polysaccharides that have a considerably high [*η*] value [44]. In addition, GBP was found to be a linear polymer exhibiting a random coil conformation, demonstrated by a Mark–Houwink–Sakurada α value of 0.702. This interpretation is based on the accepted values of α for polymers, wherein the α value of 0.5–0.8 represents a malleable random coil conformation, a α value greater than 0.8 denotes semimalleable or extra stretched chains [45], a α value of 0 indicates rigid sphere, one of 2 demonstrates stiff rod, and one of ~0.7 specifies a linear conformation [46]. The linear structure of GBP was also evident from the *Rg/Rh* value (1.57), where this ratio was ~0.775 for globular polymers, and a higher value demonstrated a correspondingly greater deviation from an orbicular to a more linearly stretched conformation [47].

A substantial AHG content (17.97%) was found in GBP. Numerous biological activities demonstrated by 3,6-anhydrogalactose–galactose (AHG–GAL) oligomers have been primarily linked with AHG content [16]. The AHG–GAL oligomers specifically exhibit immunoregulative, antioxidant, carcinostatic, apoptosis-inductive, antiallergic, hepatoprotective, α-glucosidase-inhibitive and antiinflammation effects [16,48]. Besides, AHG–GAL dimers, neoagarobiose, demonstrate moisturizing and whitening properties [49].

The surface topography of the GBP was illustrated was being rough with a tight structure, but it was not flat. Diverse polymeric chains in a linear conformation having an asymmetrical contour and size were observed (Figure 2A). AFM is an important tool for detecting the morphology of the polysaccharide that the macromolecules originated (Figure 3) [50]. Generally, the size of the polysaccharide molecular chain is about 0.1–1.0 nm [51]. The linearity and random coil conformation of GBP was obvious from the AFM micrograph. A height range of 0.12 to 0.58 nm was noted for GBP particles (average height of 0.30 ± 0.12 nm). Based on the result of the dilute polymer solution theory, the finding that GBP could be evidenced as being a linear polymer from SEM and AFM analysis.

Spectrum analyses from FT–IR established that GBP contained AHG and that it was indeed a sulfated polysaccharide (Figure 4). Alkane C–H stretching was evident at 2940 cm^−1^, and O–H stretching was observed at 3332 cm^−1^, while the absorption at 1645 cm^−1^ corresponded to polymer bound water [52]. Sulfates in GBP were represented by the absorption observed at 1372 cm^−1^ in the spectrum [53]. The presence of S=O of sulfate esters, on the other hand, was denoted by the absorption at 1249 cm^−1^ [54]. Similarly, the peaks observed at 1123 cm^−1^ and 1066 cm^−1^ could be attributed to the asymmetric stretching of glycosidic bonds [55] and C–O stretching of AHG [54], respectively. The absorption at 930 cm^−1^, instead, corresponded to the C–O–C stretching of AHG [56]. Moreover, the presence of the C6 group in β-D-galactose was confirmed by the absorption at 890 cm^−1^ [55]. The absorption at 580 cm^−1^ likewise represented O=S=O bending [55].

### 3.2. Antioxidant Studies

Superoxide anion (O_2_^−^), hydrogen peroxide (H_2_O_2_) and hydroxyl radical (HO) exemplify reactive oxygen species (ROS). These ROS, when accumulated in the body, cause oxidative stress culminating in a number of diseases [57]. Compounds extracted from natural sources, in general, and polysaccharides, in particular, are quite effective in scavenging ROS [58]. A dose-dependent scavenging efficacy was observed for GBP in DPPH, HO and ABTS assays (Figure 5). As the concentration increased from 0.125 to 2.0 mg/mL, the scavenging abilities of GBP increased. A greater scavenging capacity corresponding to 19.80%, 8.80% and 25.42% for DPPH, HO and ABTS radicals, respectively, was observed when GBP was used in the highest tested concentration of 2 mg/mL. Conversely, the lowest tested concentration of GBP (0.125 mg/mL) yielded the least observed DPPH, HO and ABTS scavenging activities (7.71%, 1.75% and 4.46%, respectively). The findings were therefore in good agreement with the electron transfer ability [59] and antioxidant effectivity of polysaccharides [60]. The antioxidant activity of polysaccharides from *G. blodgettii* might be related to the content of sulfate. Sulfated polysaccharides extracted from other red algae also possessed antioxidant activities. Di found that polysaccharides from *Gracilaria rubra* with a sulfuric radical of 12.42% presented an excellent antioxidant activity; the scavenging effects of DPPH, ABTS and superoxide anion radical were 30.67%, 55.34% and 72.82%, respectively [61]. The DPPH and hydroxyl radical activities of sulfated polysaccharides from *Gracilaria birdiae* (a sulfate content of 8.4%) were also exihibited [62].

## 4. Conclusions

The physical and chemical characterization, and ROS scavenging of GBP were studied in this work. According to the information obtained from the physical and chemical analysis, it was confirmed that GBP was in fact a sulfated polysaccharide occurring in a linear random coil conformation. This study also pointed to the presence of a considerable quantity of AHG in GBP, which is an important bioactive compound for human health. Similarly, the enormous dissimilarity in the setting and melting points observed for GBP echoes its usefulness in relation to functional food industrial processes and an improved shelf-life in hot environments. Furthermore, it is evident that sulfated polysaccharide extracted from *G. blodgettii* is a promising antioxidant substance in the functional food industry owing to its ROS scavenging efficacy.

## Figures and Tables

**Figure 1 polymers-13-00442-f001:**
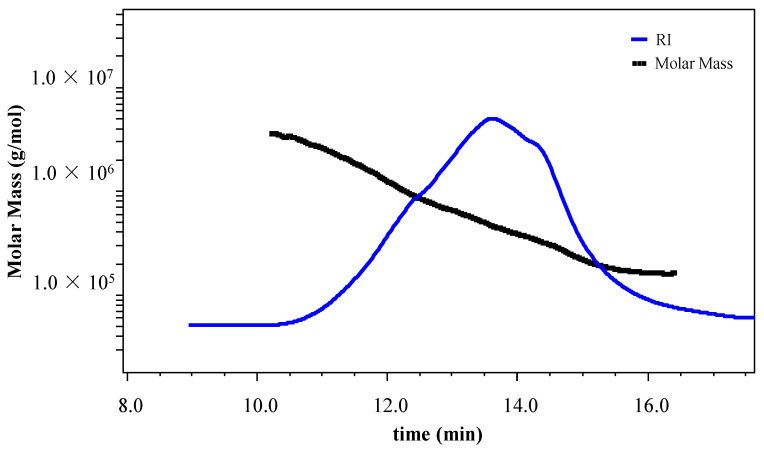
Molecular weight distribution depicted by HPSEC-Visc-MALLS-RID of *G. blodgettii* polysaccharide.

**Figure 2 polymers-13-00442-f002:**
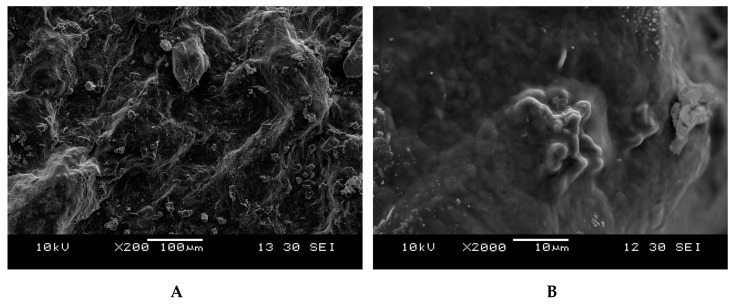
Scanning electron micrograph of *G. blodgettii* polysaccharide: magnification (**A**) 200×; (**B**) magnification 2000×.

**Figure 3 polymers-13-00442-f003:**
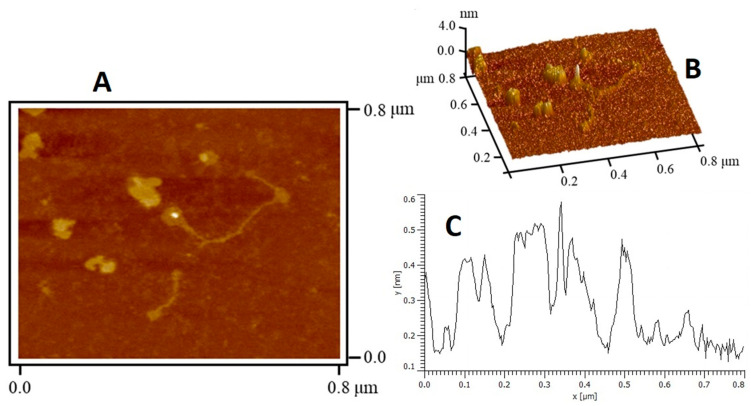
Atomic force micrograph of *G. blodgettii* polysaccharide: (**A**) 2D; (**B**) 3D; (**C**) height distribution graph.

**Figure 4 polymers-13-00442-f004:**
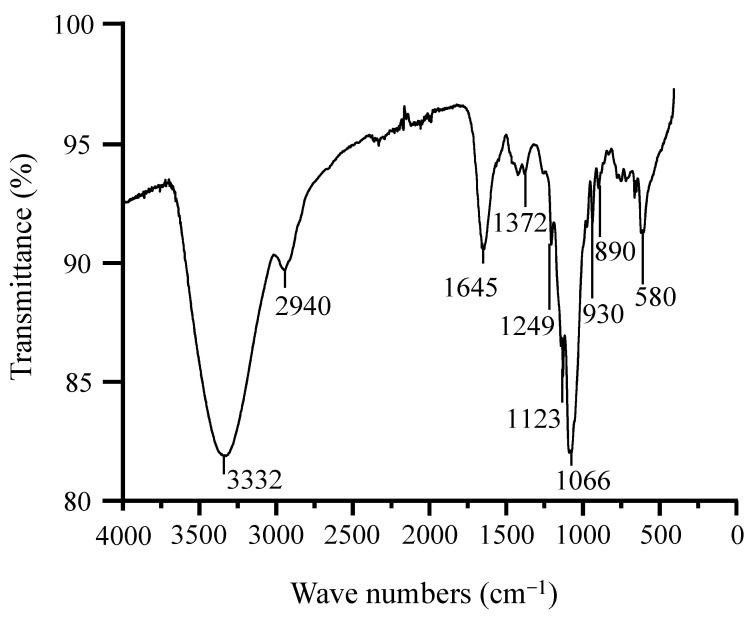
Fourier transform infrared spectra of *G. blodgettii* polysaccharide.

**Figure 5 polymers-13-00442-f005:**
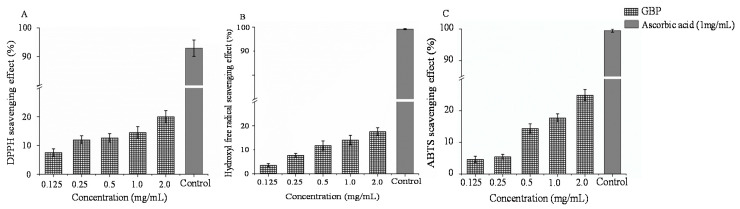
(**A**) DPPH scavenging effect, (**B**) HO scavenging effect and (**C**) ABTS scavenging effect of *G. blodgettii* polysaccharide (mean ± standard deviation, *n* = 3).

**Table 1 polymers-13-00442-t001:** Physical and chemical characteristics of *G. blodgettii* polysaccharides.

Index	Values	Index	Values
Yield (%)	24.12 ± 0.59 *	Molecular weight, *M*_n_ (Da)	4.169 × 10^5^
Sulfate content (%)	9.16 ± 0.59 *	Polydispersity (*M*_w_/*M*_n_)	1.475
Total sugar (%)	18.84 ± 0.83 *	Intrinsic viscosity (mL/g)	844.12
Setting point (°C)	27.33 ± 0.65 *	Mark–Houwink–Sakurada α	0.702
Melting point (°C)	64.33 ± 6.51 *	Hydrodynamic radius (nm)	41.763
3,6-anhydro-galactose (%)	17.97 ± 0.46 *	Radius of gyration (nm)	65.6
Molecular weight, *M*_w_ (Da)	6.149 × 10^5^		

* mean ± standard deviation, *n* = 3.

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
