# Peer review of "Antioxidant Potential of Physicochemically Characterized Gracilaria blodgettii Sulfated Polysaccharides"

_polymers, 2021, doi:10.3390/polym13030442_

Round 1

Reviewer 1 Report

This manuscript presented an interesting study about the properties of Gracilaria blodgettii. However some points listed below need to be improved.

Abstract: I suggest add wt% to the contents of sulfate and AHG.

Remove the information related to the manuscript preparation before the introduction section.

Introduction: the introduction section is too short. Please consider increase this section.

Section 2.1: please describe the conditions used to dry and powdered the plant. How is the mesh size of the powder used to polysaccharide extraction?

Section 2.3.3: the GBP solution is an aqueous GPB solution?

Section 2.3.8: please add the number of scans used in FTIR analysis.

Page 6: the authors said that “The finding that GBP is a linear polymer was also evident from SEM analysis (Fig. 2A). Diverse polymeric chains, in linear conformation, having asymmetrical contour and size were observed.” However, is not possible conclude using SEM analysis that any GBP is a linear polymer and that “Diverse polymeric chains, in linear conformation, having asymmetrical contour and size were observed”. Please rewrite this part.

Figure 2: please use “magnification” instead of “zoomed” in the figure caption.

Section 3.2: improve the discussions in this section.

Reviewer 2 Report

Dear Sirs,

The article entitled “Antioxidant potential of physicochemically characterized Gracilaria blodgettii sulfated polysaccharides” has the main goal of characterizing this polysaccharide particularly highlighting its antioxidant potential for prospective applications in food, cosmetic and pharmaceutical applications. Content is interesting but article reading is not fluid. That should be improved. Content is new, but its strength insufficiently detailed. Please compare your work to others in the literature, name the ones with DOI: 10.1016/j.foodhyd.2018.12.007, 10.3390/md17120674, 10.1016/j.carbpol.2010.04.031, 10.1590/0104-1428.013116, 10.1099/ijsem.0.000991 among others, thereby reinforcing the novelty of your article.

Abstract: in my opinion, a sentence presenting the polysaccharide and its emerging potential at the beginning of the abstract, would enrich its content. The description of the polymer’s novelty in the field could also be endowed of more fluidity. Origin, monomeric units, key properties and key bioactivity potential. It is also unclear from the abstract why the authors focused on its antioxidant activity, and how that can relate to food industry, exactly. Is it for use as food packaging materials? If so, how is the polymer better than current alternatives?

Please delete content from lines 26-32.

Line 35: “organisms [1]. Marine Rhodophytes” – organisms, marine Rhodophytes. These red algae are… ; You need also to describe what they are before identifying their main constituents. Distinctive how? That is too vague. See line 41, which identify red algae. You need to rearrange your information, otherwise it is confusing to read.

Line 46: source agar for what exactly?

Line 50-51: that are overexpressed in cancer, …

Lines 53-54: OK but how can you predict that it has antioxidant potential? Similarities in chemical composition, etc, when compared to the other polysaccharides that you mention? Can you indicate some examples such as carrageenans? And why red algae and not others? I know for instance that fucoidan and ulvans have great antioxidant potential. Comparisons between multiple red algar polysaccharides as well as other algae are lacking.

Line 57: and also in the medical field.

Personally, I would start polymer characterization with FTIR for an overview on its chemical composition. Then, quantification of sulphate and sugar content.

Line 169: thrice? For three times? Performed in triplicate? The section of statistical analysis is incomplete. Please remove text from the template and indicate which tests were done to infer on statistically significant differences.

Lines 177-178: in the past? When exactly? Please add references supporting this type of information.

Line 179: yield related to what?

Lines 181-183: “The food, cosmetics and pharmaceutical applications of sulfated polysaccharides extracted from seaweed have been in the limelight during the previous few decades [31].” – which polysaccharides have been under the light?

Lines 189-192: “Being farther apart, such setting and melting point allows for high stability amidst increased temperatures during industrial processing [12]. Moreover, higher melting point also corresponds to greater utility in hot environments and in baked stuff [32]”. – indeed, you need to clarify your intended application, otherwise it is too confusing. If it is food industry, please indicate one-time other potential applications (cosmetics, pharmaceutics, etc.) and then focus, and detail, aspects related to the food industry.

Lines 255-256: “These ROS, when accumulated in the body, cause oxidative stress culminating in a number of diseases [48].” – though the intended application of your polymer, how is that relevant? In which circumstances would the polymer get into contact with the human body, so that its bioactivity could take place thereby counteracting any particular oxidative stress?

Line 275: a linear random coil conformation? SEM and AFM are insufficient to make this assumption… either you do a proper analysis of GBP chain conformation, or you would talk about appearances and hints.

Line 276: important for what?

Line 278: which industrial processes?... the whole manuscript is like this, too vague. This really needs to be improved…

In my opinion, the conclusion needs to be finalized stating indeed that the polymer has promising antioxidant properties that can be useful in multiple fields (such as the one that you should further highlight) but also detach its potential from the one of similar polysaccharides. Otherwise, it can be interpreted as just another algae-derived sulphated polysaccharide with some antioxidant potential.

Reviewer 3 Report

The paper is focused on the physicochemical characterization of selected polysaccharides. Proposed work is interesting and designed adequately. The topic selected by Authors fits well into the currently observed trends based on the search for the natural compounds with beneficial properties and application potential in medicine and the relative areas. In general, the article may be considered for publication but some improvements are suggested, i.e.:

  • Section „0” should be removed.
  • Section 2.4.: Authors proposed few analyses aimed at the determining the antioxidant potential of GBP. Brief explanation of the meaningless of these investigations (i.e. the determining of each scavengings activity) needs to be added by Authors.
  • Figure 4: firstly, the description “Transmittance” will be enough in the case of y axis. Next, the range of x axis should be as follows: 4000 – 500 cm-1.
  • Quality of figures needs to be improved.
  • From the editorial point of view, Authors should remember about subscripts when writing the chemical formulas (for example when writing the formula of the hydrogen peroxide).
  • Section References should be prepared according to the requirements of the journal, i.e. the whole journal names should be replaced by their abbreviations.
